# An Overview of Deep-Learning-Based Methods for Cardiovascular Risk Assessment with Retinal Images

**DOI:** 10.3390/diagnostics13010068

**Published:** 2022-12-26

**Authors:** Rubén G. Barriada, David Masip

**Affiliations:** AIWell Research Group, Faculty of Computer Science, Multimedia and Telecommunications, Universitat Oberta de Catalunya, 08018 Barcelona, Spain

**Keywords:** healthcare, artificial intelligence, deep learning, medical imaging, retinal fundus image, retinal photography analysis, oculomics, convolutional neural networks, cardiovascular diseases

## Abstract

Cardiovascular diseases (CVDs) are one of the most prevalent causes of premature death. Early detection is crucial to prevent and address CVDs in a timely manner. Recent advances in oculomics show that retina fundus imaging (RFI) can carry relevant information for the early diagnosis of several systemic diseases. There is a large corpus of RFI systematically acquired for diagnosing eye-related diseases that could be used for CVDs prevention. Nevertheless, public health systems cannot afford to dedicate expert physicians to only deal with this data, posing the need for automated diagnosis tools that can raise alarms for patients at risk. Artificial Intelligence (AI) and, particularly, deep learning models, became a strong alternative to provide computerized pre-diagnosis for patient risk retrieval. This paper provides a novel review of the major achievements of the recent state-of-the-art DL approaches to automated CVDs diagnosis. This overview gathers commonly used datasets, pre-processing techniques, evaluation metrics and deep learning approaches used in 30 different studies. Based on the reviewed articles, this work proposes a classification taxonomy depending on the prediction target and summarizes future research challenges that have to be tackled to progress in this line.

## 1. Introduction

Artificial Intelligence (AI) models have been playing an increasing role in medical research imaging in the last two decades. For diagnostic imaging alone, the number of publications on AI has increased from about 100–150 per year in 2007–2008 to 1000–1100 per year in 2017–2018 [1]. The progress of computing systems during the last years has allowed deep learning (DL), a sub-field of AI, to become a feasible methodology to analyze complex sources of data, such as medical images. There are multiple applications where deep learning has achieved impressive results: mammography mass classification [2], brain lesion segmentation [3], skin lesion classification [4], COVID-19 prediction [5], etc. The authors in [6] provided a wide review covering the main architectures, techniques and applications of DL applied to medical image analysis.

Within medical imaging techniques, retinal photography analysis has gained popularity due to its noninvasive and cost-effective nature [7]. Retinal fundus images (RFI) are obtained from the projection of the rear part of the eye (fundus) onto a 2D plane using a monocular camera. Different biomarkers and eye structures can be identified from a RFI, playing an important role in identifying retinal abnormalities and diseases, such as glaucoma, diabetic retinopathy (DR), macular edema degeneration, etc. In recent years, deep learning applied to oculomics has aroused great interest in the scientific community. Studies on the identification and prediction of ocular biomarkers of systemic diseases are becoming increasingly interesting for researchers in the field [8]. Deep learning techniques are providing insights about eye–body associations through retinal morphology analysis to enhance the understanding of complex disorders, such as musculoskeletal diseases [9], traumatic brain injury [10], cardiovascular disease [11], renal impairment [12], Alzheimer’s disease [13] or anemia detection [14].

Cardiovascular diseases (CVDs) remain a leading cause of death globally [15]. According to the World Health Organization, an estimated 17.9 million people died from CVDs in 2019, representing 32% of all global deaths. According to this report [16], in 2015, 15.2 million deaths were produced by stroke and ischemic heart disease solely, 85.1% of the total deaths provoked by cardiovascular events. Most cardiovascular diseases can be prevented by addressing risk factors, both individual, such as chronological age, gender, smoking status, blood pressure and body mass index (BMI), or metabolic, such as glucose and cholesterol levels [17]. Detecting CVDs as early as possible is critical for efficient clinical treatment, and this is where deep learning models can be incorporated into the diagnostic process. The motivation behind this paper is focused on surveying the main contributions of this automated diagnosis of CVDs from RFI.

The proposed overview aims to provide a glance into the current state-of-the-art of DL strategies to assess cardiovascular diseases by analyzing retinal fundus images. Moreover, this study intends to highlight the main obstacles to face in these applications and possible future work to be performed for new DL-based research methods for CVDs diagnosis. Section 2 covers the domain knowledge regarding the review topic, analyzing cardiovascular risk and delving into deep learning models and retinal fundus image structures. Analyzed materials and methods are reviewed in Section 3, where commonly used datasets, pre-processing techniques and metrics are explained. Section 4 gathers the literature review of selected studies along with comparisons on the automated diagnosis of cardiopathies. Finally, the review results and conclusions are presented in Section 5.

## 2. Domain Knowledge

### 2.1. Classification of Cardiovascular Diseases

In essence, cardiovascular diseases can be defined as a set of conditions that may affect the structures or behavior of the heart and vascular system, causing malfunction and even death. There are different types of CVDs: (https://www.webmd.com/heart-disease/guide/diseases-cardiovascular) Accessed on 25 November 2022.

**Arrhythmias:** Irregular or abnormal heartbeat that can bring on an uneven heartbeat or a heartbeat that is either too slow or too fast.**Aorta Disease and Marfan Syndrome:** This disease is produced when the aorta walls are weak. This can put extra stress on the aorta, which increases the risk of a deadly dissection or rupture.**Cardiomyopathies:** Diseases related to the heart muscle when it is unusually big, thick or stiff. The heart cannot pump blood as well as it should.**Congenital Heart Disease:** Abnormalities in one or more parts of the heart or blood vessels before birth that may appear for different reasons: genetics, virus, alcohol or drug exposure during pregnancy.**Coronary Artery Disease:** Produced when plaque builds up and hardens in the arteries that provide the heart vital oxygen and nutrients. That hardening is also called atherosclerosis.**Deep Vein Thrombosis and Pulmonary Embolism**: When blood clots, normally formed in deep veins, such as the legs, can move in the blood flow to the lungs, provoking blocked points in the bloodstream.**Heart Failure:** It is produced when the heart does not pump as strongly as it should and may provoke swelling and shortness of breath.**Heart Valve Disease:** The valves are located at the exit of each of the four heart chambers. They keep blood flowing through the heart. Examples of heart valve problems include:-**Aortic stenosis:** The aortic valve narrows. It slows blood flow from the heart to the rest of the body.-**Mitral valve insufficiency:** Caused by a malfunction in the mitral valve that may end up in a lung fluid backup due to blood leaking.-**Mitral valve prolapse:** The mitral valve does not close correctly between the left upper and left lower chambers.-**Pericarditis:** Often provoked by an infection, the lining around the heart is inflamed.-**Rheumatic Heart Disease:** This condition is most common in children. The heart valves are damaged due to rheumatic fever, causing an inflammatory disease.-**Stroke:** Reduction or block of the blood to the brain, depriving the correct contribution of oxygen and nutrients. For instance, a blocked artery or a leaking blood vessel can lead to a stroke event.**Peripheral vascular disease:** Involving any abnormality that directly alters the circulatory system, e.g., leg artery diseases may affect blood flow to the brain, ending up in a stroke.

The list of risk factors for having a cardiovascular event is extensive: high blood pressure, smoking, high cholesterol, diabetes, inactivity, being overweight or obese, family history of CVD, etc. (https://www.nhs.uk/conditions/cardiovascular-disease/) Accessed on 25 November 2022.

### 2.2. Deep Learning Approach

Deep learning techniques have been gradually introduced in several fields, including bioinformatics, the domains of which comprise branches such as omics, biomedical signal processing and medical imaging.

Deep learning methods are based on artificial neural networks, which are slightly inspired by biological neural organization, where the neurons are processing units organized in connected layers. These computational structures learn how to perform certain tasks just by considering a relatively large set of input examples without being specifically designed for the task. Their generalization capabilities are input-dependent, meaning that with the same network structures, the learned task might be different if the input example set is different. Specifically, deep learning uses deep neural network concepts where the approach is based on layer specialization. Each network layer gathers concrete knowledge that is afterward shared among specific layers to impulse the general learning process. This general idea generates very different network architectures, such as deep belief networks (DBN), recurrent neural networks (RNN), convolutional neural networks (CNN) (example in Figure 1), etc., which are specially oriented to specific problem solving: classification, segmentation, prediction, etc.

Depending on the context, different deep learning strategies can be followed:**Supervised learning:** The quality of deep neural network performance is strongly influenced by the number of labeled/supervised images. The more images are in the training dataset, the higher the accuracy achieved by the model. To solve the problem of a lack of input data, a commonly used option is transfer learning. This approach tackles the small training size problem by pre-training the model using different natural examples, based on the premise that first network layers learn similar features and the later layers are the problem-specialized ones.**Unsupervised learning:** The model learns common associations and structures within the input set. Sometimes there is access to a large dataset of unlabeled data that can be exploited in a semi-supervised or self-supervised way. The main idea is to use these data during the training process to increase the model’s robustness, sometimes even surpassing the supervised cases [18].**Semi-supervised learning:** The data used to perform certain learning tasks are both labeled and unlabeled. It typically incorporates a small size of labeled data in combination with large amounts of unlabeled data. The difficulties here are that unlabeled data can be exploited if they provide information relevant to label estimation that is not present in the labeled data or cannot be easily obtained. This learning method requires the information to be determinable [19].

**Figure 1 diagnostics-13-00068-f001:**
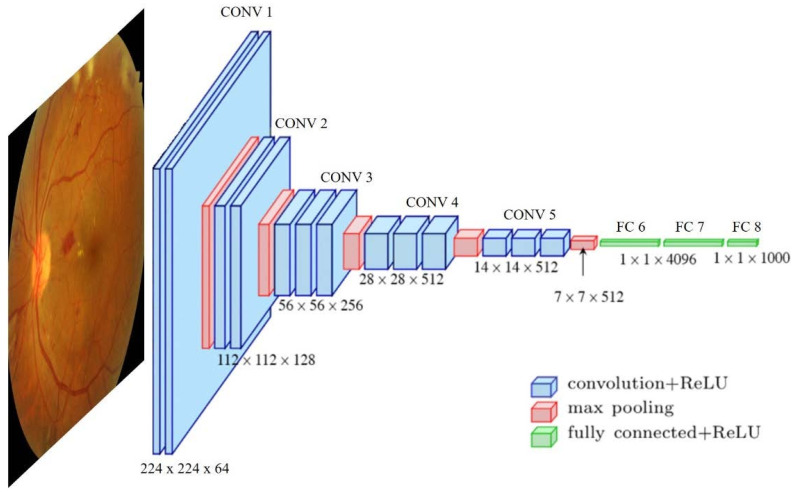
Example of a commonly used convolutional neural network used in medical image analysis. VGG16 architecture proposed by [20] and applied to task classification. Image source [7].

### 2.3. Retinal Fundus Images

Researchers and practitioners have realized the many possibilities offered by retinal imaging and its ability to provide information on retinal vasculature. Moreover, it is the only internal vascular system that can be observed in a noninvasive condition of the human body. Disease-related information can be extracted from fundus images. Thus, RFIs have been used in the medical field to reveal many important systemic diseases of the human body that cause specific reactions in the retina.

A RFI is a projection of the fundus captured by a monocular camera on a 2D plane. These images make it possible to detect visible abnormalities and lesions in the eye quickly and with little associated cost. For instance, features such as exudates, microaneurysms and hemorrhages are visible pathological signs of diabetic retinopathy. An example of a retinal fundus image with marked features is shown in Figure 2.

## 3. Materials and Methods

### 3.1. Article Search and Selection Strategy

The overall search approach was based on data-mining terms from scientific databases. Concretely, we used GoogleScholar (https://scholar.google.es) and PubMed (https://pubmed.ncbi.nlm.nih.gov/), accessed from 25 October to 10 November 2022. In both databases, the same search criteria were applied: open-access articles written in English from 2018. The search was performed from the exact terms: “deep learning”, “retinal images”, “cardiovascular diseases”.

The records excluded surveys or reviews. Only articles and proceeding papers were accepted. From them, only the topic relevant were kept by exploratory inspection. After the filtering process, this literature review includes 30 studies that will be detailed in the following sections. The proposed methodology can be seen in Figure 3.

### 3.2. Datasets

This section details an overview of widely used datasets in deep learning applications. Concretely, the datasets used in the studies of this review are listed in Table 1. The table lists out datasets, the number of images contained, image size, label description and the main task they are used for. Only public datasets and datasets with restricted distribution (via request) are mentioned in the table.

The rest of the datasets used in the research articles are shown in Section 4, are private domain datasets from clinic studies and hospitals, and their detailed features are not available.

### 3.3. Pre-Processing Techniques

Within medical imaging, the RFI area is quite controlled and not so exposed to ambient noise, so they are usually inputs of high quality. However, fundus images are usually pre-processed to make more robust models and better predictions. Figure 4 shows examples of the main pre-processing operations used in the reviewed studies, having the original Figure 4a as a reference.

Within image color transformations, grayscale conversion (Figure 4b) is one the most common operations used when trying to reduce the number of input channels and thus reduce the computational load in the architecture. Moreover, in some DL models, having a single color input increases the contrast and produces improvements in the model performance.

Contrast enhancement is also another commonly used technique to emphasize the main image features. Among several methods, Contrast-Limited Adaptive Histogram Equalization (CLAHE) stands out as one of the most widely used. It was first proposed in [34]. It is an easy-to-implement technique that addresses global histogram equalization problems using local image statistics. This technique is usually used in offline steps due to its expensive computational cost. However, its use enhances interesting retinal features, such as microaneurysms and small hemorrhages.

Noise reduction is also an extended technique applied in DL to increase image quality in the sense of decreased abnormal signals within the image that may slow down the training process. Among the known methods, Gaussian and median filtering are the most commonly used in the literature reviewed. The median filter is used to preserve useful details in the image, usually edges. It considers each pixel in the image, sorting them using a NxN mask. It replaces the pixel value with the median of the values covered by the mask. An example of this method can be seen in Figure 4f. The Gaussian filter also is applied to reduce detail and image noise. The output is the result of blurring an input image by means of a Gaussian function applied at the pixel level.

Normalization is used to change the range of the pixel intensity value. It can also be called contrast stretching. Min-Max Normalization scales and translates the values of the image so that they are linearly mapped into the interval between [α,β] (inclusive), where α and β are the lower and upper range boundaries, respectively. An example can be seen in Figure 4e.

When images are too light or too dark, Gamma correction is a method that allows the brightness of an image to be adjusted. The Gamma-adjustment can be expressed as defined in Equation (Equation 1):(1)$$O={\left(\frac{I}{255}\right)}^{\frac{1}{\gamma }}\cdot 255,\;\mathrm{where}\;\gamma \left\{\begin{array}{c} \mathrm{if}\;\gamma <1:\mathrm{darker}\;\mathrm{image}\\ \mathrm{if}\;\gamma >1:\mathrm{lighter}\;\mathrm{image}\\ \mathrm{if}\;\gamma =1:\mathrm{no}\;\mathrm{effect} \end{array}\right.$$
where *O* is the output (pixel value [0,255]), *I* is the input image (pixel value [0,255]) and γ is the gamma factor. An example of the Gamma adjustment can be seen in Figure 4d.

Data augmentation is a mechanism that consists of physically manipulating a sample image by applying image transformations to produce more inputs. This technique prevents overfitting during the training phase under a lack of input samples. The hypothesis here is that a greater amount of information can be learned by the network through augmentations from the original images [35]. Classical augmentation operations are random cropping, horizontal and vertical flipping, and Red-Green-Blue (RGB) intensity channel modification, to name a few. Examples of data augmentation are shown in Figure 4g–i.

The straight use of RFI in DL models increases the computational cost due to the high-resolution characteristic, especially in very deep architectures. For this reason, in most of the studies, images are usually scaled (resized) before being introduced into the model, always trying to maintain the image aspect ratio.

### 3.4. Evaluation Metrics

Deep learning algorithms should incorporate evaluation methods so the model performance can be evaluated and measured. Table 2 lists the metrics used in the reviewed articles. Commonly, the evaluation metric selected depends on the deep-learning task type. For instance, metrics such as accuracy, specificity sensitivity or Area Under Receiver Operating Characteristic are used as performance indicators for a classification application and the Dice index is normally used to measure the segmentation performance in RFI.

## 4. Automated Diagnosis of Cardiopathies

Descriptions of the studies reviewed are provided in this section. Broadly speaking, DL methods are able to extract patterns of information from RFI. During the training process, a group of images from the dataset is fed as input to the model. The information contained in the image is computationally decomposed and analyzed through the layers of network architecture until it reaches the network output. In the initial layers, basic structures within the image, such as edges or colors, are learned. The deeper layers of the network are dedicated to learning more complex retinal information, such as veins, exudates, micro-hemorrhages, etc. In the case of supervised learning, the error in the output is computed with the label associated with the input image and back propagated to the initial layers by adjusting the weights that regulate each layer. After the training phase, the rest of the dataset is used to evaluate the generalization capability of the network. The quantity and quality of the input dataset are determinants for the accuracy capability of the network. This is why the pre-processing operations we saw in Section 3.3 are so important. To increase the number of input images (data augmentation) or to provide a cleaner input image through noise reduction or contrast enhancement processes will make the network learn the characteristics of each image better.

The cardiovascular damage diagnosis approach is different in each article, varying not only in the clinical analysis approach but also in the deep learning techniques, datasets, strategies, etc. Therefore, we propose a classification of the articles in the following subsets:Extracting biomarkers: Methods oriented to cardiovascular anomalies detection but only focused on retinal biomarker extraction.Prediction risk factors: Approaches to predict risk factors at the individual (chronological age) or metabolic (coronary artery calcium, hypertension) levels that may result in cardiovascular damage.Prediction of cardiovascular events: Direct cardiovascular events (stroke, ictus).

### 4.1. Automated Methods for Extracting Biomarkers

Relevant deep-learning approaches regarding vascular feature segmentation are detailed in this section. The quantitative measurement of retinal vessels is important for the diagnosis, prevention and therapeutic evaluation of cardiovascular system-related diseases. Retinal vessels are composed of arteries, arterioles, venules and capillaries. Certain abnormalities in vessel function and geometry, such as vasoconstriction, narrowing and refraction of small arteries and arterioles, have been related to cardiovascular events (left ventricular failure, stroke) and nephropathy, such as hypertension [36]. These retinal vascular changes can be measured from the point of view of different parameters, as can be seen in Table 3.

A proposal for a deep-learning method to quantify retinal microvasculature and vessel segmentation is given in [37]. They extended a U-net architecture into multiple branches in order to simultaneously segment the vein, artery and optic disc. The U-net architecture was proposed in [38] and is based on a symmetrical encoder and decoder structure used for image segmentation. The first is responsible for extracting features from input images, while the decoder reconstructs the images for the final output. The performance of the model achieved an AUC of over 90% for both vein and artery segmentation in different datasets.

Motivated by the challenging problems when segmenting coronary arteries, [39] tries to mitigate the low performance in classic unsupervised methods and the time-consuming need for manual annotation. They propose a transfer learning approach based on Generative Adversarial Networks (GAN) [40]. A GAN architecture is based on two neural networks that compete with each other to be more accurate in their predictions. They run unsupervised and use a zero-sum cooperative game framework to learn. After training the GAN mode for coronary artery segmentation, they were also able to transfer the knowledge to an unlabeled digital subtraction angiography (DSA) dataset by using a U-Net architecture. The GAN-proposed network reported an accuracy of 0.953 compared to a classical U-Net performance of 0.921.

Another contribution of retinal vessel segmentation is described in [41]. This paper proposes a complex model based on U-net and an attention mechanism, which through this mechanism, the network can recalibrate the features, selectively emphasize the useful features and suppress the bad ones. The model was able to report over 0.98% AUC in the DRIVE and STARE datasets.

Further, also based on U-Net architecture, paper [42] proposed a method to measure vascular branching complexity using an ensemble model of U-Nets to segment the microvasculature and thus calculate vascular density and fractal dimension (FD). On the test set, the model achieved an 82.1% Dice similarity coefficient, 97.4% pixel-wise accuracy, 0.99% AUC for FD and 0.88 for vascular density.

A contribution to retinal vessel detection has been made in [43]. The main motivation is to have available retinal microvasculature for further analysis, such as vessel diameter and bifurcation angle quantification. They propose a custom implementation called Faster Region-based Convolutional Neural Network (Faster-RCNN). Briefly, this architecture is composed of three modules: A feature network to generate feature maps from the input image. A separately trained network, Region Proposal Network (RPN), generates bounding boxes that contain different features or objects extracted from feature maps, and a Detection Network, which takes input from both the RPN and feature network to detect the expected features. They report a capability of extracting the true vessels of the retina with a sensitivity of 92.81% and 62.34% Positive Predictive Value (PPV).

A list of the main characteristics of the reviewed methods is shown in Table 4.

### 4.2. Automated Prediction of Cardiovascular Risk Factors

There are certain health conditions, not only at the metabolic level but also at the individual level, such as age or lifestyle, that are indicators of cardiovascular risk. These biomarkers are considered a proxy and are essential in the diagnosis of cardiovascular disease. One of the most successful predictors of cardiopathies is the presence and degree of DR. Recent studies show that age (p<0.002), gender (p<0.039) and DR (p<0.00001) are significantly different in patients with a high CAC score (≥400, considered the proper predictive tool according to the American College of Cardiology [44]) with respect to patients with a CAC score below 400 [45].

Consequently, one of the first proxies used for assessing CVD with deep learning was diabetic retinopathy disease prediction. It often appears from type 1 and 2 diabetes complications due to retinal blood vessel deterioration and might be a potential risk factor.

In [46], the authors applied a deep learning model based on the InceptionV3 architecture [47] for the detection of DR and also diabetic macular edema reporting an AUC of 0.991 tested on EyePACS and 0.990 in MESSIDOR-2. Later on, in [48], the authors proposed a hybrid model combining a custom CNN model for future extraction feeding a decision tree classification model [49] to predict DR. The classification method discriminated between healthy fundus images and having DR, identifying relevant cases for medical referral. They reported testing results on the MESSIDOR-2 and E-Ophtha databases of 0.94 and 0.95 AUC scores, respectively. They used heat maps to provide what areas in the image influenced the model to produce the output. Moreover, in study [50], the authors proposed a custom CNN model for binary classification (YES/NO) to predict DR from RFI, where they achieved an accuracy of 89%. Another contribution in article [51] proposed a deep learning method to detect referable/vision-threatening DR, in addition to possible glaucoma and age-related macular degeneration (AMD). By adapting a Visual Geometry Group (VGG) architecture [20], they report an AUC of referable DR of 0.93%. For vision-threatening DR, the AUC was 0.958%; for possible glaucoma, the AUC was 0.942%; and for AMD, the AUC was 0.931%. Focused on DR screening and other eye-related diseases, the work proposed in paper [52] trained a deep learning algorithm pointing out the multi-ethnic nature of the patients contributing to all the datasets used during training and validation. The architecture used consisted of eight modified variants of the VGG-19 CNN, two for DR, two for AMD, two for glaucoma, one for assessing quality images and one for rejecting invalid non-retinal images. They report over 0.93% AUC for any retinal disease. In the work proposed in article [53], the authors combined the use of a U-Net for semantic segmentation of the blood vessels and a deep residual network (ResNet-101 [54]) for severity classification on both vascular and full images. Vessel reconstruction through harmonic descriptors is also used as a smoothing and de-noising tool. They report that at least 93.8% of DR (No-Refer vs. Refer) classification can be related to vasculature defects. In [55], the authors implement a DL method to asses Retinopathy of Prematurity (ROP), being a leading cause of child vision loss and possible future cardiovascular complications. They established an ROP scale of 1-9 to score the retinal vascular abnormality, reporting an AUC of 0.96% for ROP-1 on the test set. They use two DL models in a row: the first a U-net to segment the vessels over the original RFI. The second one to classify disease severity by an InceptionV1 net [56].

Retinal change detection is useful for predicting biomarkers related to cardiovascular and chronic disease. The evaluated studies have shown that retinal photography-based deep-learning methods can be implemented for biomarker estimation. Poplin and colleagues proposed a DL model to predict cardiovascular risk factors with reasonable accuracy: age (within 3.26 years), gender (0.97 AUC), smoking status (0.71 AUC), HbA1c (within 1.39%) and systolic blood pressure (within 11.23 mmHg) [57]. Given the good results, they tried to also predict future major cardiac events (within 5 years) with an AUC of 0.70. They draw attention maps for each risk factor to identify the anatomical regions that the algorithm might have been using to make its predictions. Similar studies were reported in paper [58]. They use a deep learning model to predict cardiometabolic risk factors: age, sex, blood pressure, HbA1c, lipid panel, sex steroid hormones and bioimpedance measurements. The architecture proposed was MobileNet-V2 [59], known to be a light and fast DL model boosted by transfer learning with ImageNet [60]. Another contribution was made in [61], where the authors proposed a DL method to measure the retinal-vessel caliber with RFI. The model achieved comparable estimations with expert practitioners relating vessel caliber and CVD evidence, including biomarkers such as BMI, blood pressure, glycated hemoglobin and total cholesterol. Deep learning model measurements agreed with high similarity with experts having correlation coefficients between 0.82 and 0.95.

A DL-based biomarker predictor model is proposed in article [62]. They independently trained 47 VGG16 models to predict 47 systemic biomarkers: demographic factors (age and sex), relevance to CVD (blood pressure, body composition, renal function, lipid profile, diabetes-related measures and C-reactive protein), the predictable capability from hematological parameters and blood data, such as biochemical, liver and thyroid markers. Moreover, they used saliency maps to provide algorithm attention information.

When it comes to cardiovascular disease, age is undoubtedly a factor to be taken into account. The work in [63] contributed to predicting biological age from RFI and evaluated the performance of this marker in the risk stratification of mortality and major morbidity in general populations. They used a VGG classifier to implement this approach to measure aging with experimental results (c-index = 0.70, sensitivity = 0.76, specificity = 0.55). Their analysis includes saliency maps to provide regions of model attention. In the same line, in article [64], the authors proposed the use of retinal age gap as a predictive biomarker (predicted - chronological age) for CVD using Xception [65] implementation, reporting a correlation of 0.80 (*p* < 0.001) and a mean absolute error (MSA) of 3.55. Additionally, they related RAG to regression models with arterial stiffness and incident CVD, ensuring an increased risk when age reached 1.21.

Past studies showed that coronary artery calcium (CAC) had a low ability to predict cardiovascular events [66]. In practice, it could be a good predictor, but these study showed that 2% of the sampled population had a cardiovascular event, and one-third of the middle-aged and 100% of the older individuals of the total population had coronary calcification. However, later studies claim CAC scoring is a significant method for predicting cardiovascular events [67], especially among individuals without diabetes [45]. To mitigate the presence of coronary calcification in a large part of the population, CAC score stratification is performed by Agatston units (AU), where a CAC score lower than 100 AU involves low risk, between 100 and 400 AU is moderate risk and greater than 400 AU is high risk. Automated prediction DL-based methods using RFI are of special interest since CAC score measurement needs the use of Computed Tomography (CT) scans, which are expensive and involve radiation risks. These methods intend to predict cardiovascular risk by stratifying the CAC score.

The work described in [68] used InceptionV3 architecture to evaluate the high accumulation of CAC using RFI. Fundus images and CAC scans were taken on the same day. They discriminate no CAC vs. CAC>100 with an AUC of 82.3% and 83.2% using unilateral and bilateral RFI, respectively. They also used a setting combining DL prediction with other risk factors, such as age, gender and hypertension, and combined them into a regression model to increase the prediction. They tested the algorithm with different inputs: fovea inpainted, vessels inpainted, unilateral RFI and bilateral RFI, where the last one provided better results. Heat maps are provided to show areas of interest in the input data. The RetiCAC framework was proposed in [69]. They implement a DL method to predict the presence of CAC from fundus images. They found that the CAC score assessment model performed better than the prediction of other risk factors alone (AUC 0.742). From here, they proposed a CV risk stratification system with comparable performance to a CT scan: RetiCAC score (based on a probability score derived from the DL model). Another contribution of CAC assessment is detailed in article [70]. The authors proposed an automated hybrid method to predict, from fundus images, whether the CAC score surpasses a threshold set to 400 defined by experts. They defined a pipeline combining independent results from both a VGG16 model (trained on RFI) and classic machine learning classifiers (trained on clinical data: age and presence of DR) to predict (CAC < 400/CAC > 400). They reported complementary results, proposing two applications that can benefit from the combination of image analysis and clinical data: an application for clinical diagnosis (75% Recall) and an application for image retrieval of large databases (91% Precision).

Abnormalities of the retinal vasculature may reflect the degree of microvascular damage due to hypertension, atherosclerosis or both, which may end up in cerebrovascular and cardiovascular complications [71]. With this motivation, the authors of [72] proposed a prediction model to evaluate biomarkers, such as hypertension, hyperglycemia and dyslipidemia. They trained an InceptionV3 architecture achieving promising results: an AUC of 0.88 for predicting hyperglycemia, of 0.766 for predicting hypertension and of 0.703 for predicting dyslipidemia. Moreover, they also trained the network to predict other risk factors (age, gender, drinking/smoking habits, BMI, etc.) directly related to CVD, reporting AUCs over 0.68 in all of them. Another contribution predicting hypertensive patients was proposed in [73]. They implemented a custom deep learning architecture called Deep Neuro-Fuzzy network (DNFN), where the input data are based on a feature vector previously extracted from the RFI images. The structure of the DNFN is based on two stages: in the initial one, a deep neural network where the input and hidden layers are in charge of learning the output layer for classifying, and in the second stage, where a fuzzy logic optimization process computes the system objective. The classification accuracy reported by the model was 91.6%.

Coronary artery disease, also known as atherosclerosis, was used in article [74] as a biomarker related to CVD. The purpose of this study was to develop a deep learning model based on Xception architecture, which predicted atherosclerosis by using RFI. The model was validated in two phases: First, participants with RFI plus carotid artery sonography were used to train the deep model for the prediction of atherosclerosis. Predictions are independently made with one RFI at a time. A custom DL-FAS metric was obtained from the final averaged prediction on each eye. DL-FAS was used for validation if future CVDs can be predicted from subjects with only RFI (carotid artery sonography unavailable). The final results showed an AUC of 0.713 and an accuracy of 0.583. Attention maps are provided to reflect the main interest image region of the model. Later on, a coronary artery disease prediction model was developed in article [75]. They use retinal vascular biomarkers to predict coronary artery disease using CAD-RADS as a proxy for cardiovascular disease. They do not use the RFI to directly feed the network. Instead, they extract features from the pre-processing stage that were the inputs of the model as a feature vector. They compare the performance of the net over traditional machine learning (ML) methods outperforming the results, obtaining above AUC 0.692 in all the cases. Table 5 lists the above-mentioned works.

### 4.3. Automated Prediction of Cardiovascular Events

Retinal fundus photography has been proposed for stroke risk assessment due to its similarity between retinal and cerebral microcirculations [76]. A binary DL-based classification method was developed to predict stroke event risk, achieving the best performance with AUC ≥ 0.966. From a previous RFI pre-processing process, they fed the model with two different input images: A templated image based on contrast normalization and median-filtering transformations and a vessel image obtained from a U-Net segmentation model. VGG19 architecture was used as a classification method. They also provide heat maps with the predictions.

Stroke prediction has been another application of retinal image analysis with deep learning algorithms. The authors of paper [77] proposed an Inception-Resnet-v2 [78] to predict 10-year ischemic cardiovascular diseases (ICVD) from a Chinese population dataset. The algorithm was able to achieve an AUC of 0.971 and 0.976 in internal validation and 0.859 and 0.876 in external validation.

In [79], the authors implemented an ensemble-based framework, the architecture of which was composed of a Generative Adversarial Network (GAN) that uses a U-Net model as a generator to synthesize the images with high resolution. Afterward, an IncepcionV3 model was applied to predict the severity level of the CVD. The results show that an ensemble classifier with a CNN model had the best performance, with an improved accuracy of 91% for the different types of heart disease.

Multimodal approaches to predict CVD are also proposed. The work described in [80] used a combination of source data integrating information from RFI and dual-energy X-ray absorptiometry (DXA), demonstrating the improved use of combined information. A DL-based technique based on ResNet architecture was used to distinguish the CVD group from the control group with 75.6% accuracy. Independently, classical machine learning classifiers achieved 77.4% accuracy on DXA data. The combination of both classifiers plus a custom CNN achieved 78.3% accuracy.

Another deep-learning pipeline for CVD prediction was proposed in [81]. The study presents a hybrid system that estimates cardiac indices, such as left ventricular mass (LVM) and left ventricular end-diastolic volume (LVEDV), and predicts future myocardial infarction events. The system is composed of two main components: a multichannel variational autoencoder (mcVAE) [82] and a ResNet architecture. First, the mcVAE is designed with two pairs of encoders/decoders that train the network from RFI and cardiac magnetic resonance (CMR) with a shared latent space. Second, the learned latent space is used to train the ResNet model from CMR images reconstructed from the retinal images plus the demographic data (age, gender, HbA1c, systolic and diastolic blood pressure, smoking/alcohol habits, glucose and BMI) to estimate LVM and LVEDV. Finally, they predict the myocardial infarction risk using logistic regression with 0.80 AUC, 0.74 sensitivity and 0.71 specificity.

In the table below (Table 6), the list of reviewed methods is shown.

## 5. Conclusions

In this review, 30 recent works on the application of deep learning models to retinal images for the prediction of cardiovascular disease have been described. The main paper’s contributions are an updated (to the best of our knowledge) literature survey of the main methods to automatically predict CVDs from RFI images, a review of the main datasets used (with a specific focus on the main features of the publicly available ones) and a summary of the available experimental results. We provide a taxonomy of these methods depending on the target approach. Specifically. in the last two years, the number of publications on the subject has been prolific, which proves the general interest of the research community in the subject. Already, successful deep learning architectures in other fields are being migrated and applied to medical image analysis, and medical and deep learning practitioners are joining efforts, which is increasing the number of studies with promising results. The reviewed methods use very different data sources and strategies to reach the diagnosis of CVD. Therefore, it is difficult to compare the performance of each method from an objective point of view.

There are still classical deep learning obstacles to avoid. Future challenges will deal with data availability and model improvements. Even though some of the works reviewed here had considerably large image datasets, data accessibility is still very limited. The results in some of the works can be significantly improved by gathering more clinical data (increasing the number of patients). Along the same line, there is sometimes a lack of reproducibility in studies, which slows progress in similar research. Moreover, there is a trade-off between computing consumption and image resolution; therefore, all images are resized. Higher resolution in relevant discriminating information within retinal images (thickness, dimension, tortuosity of the vessels, etc.) would increase the model specialization capability. Finally, the black-box nature of the learning task involves difficulty in clearly understanding which parts of the image influenced the predictions of the networks. Applying deep learning models in real clinical settings would require better explainable models attached to deep learning algorithms to provide enough evidence of its decision-making process. It is necessary to isolate other pathological factors that may be influencing the decisions of DL methods in the diagnosis of cardiovascular events, such as diabetic patients. Moreover, it is desirable that automated diagnosis systems could provide an easily interpretable explanation behind their predictions that clinicians can understand and trust.

Finally, the methods reviewed here are able to predict, in the best of cases, CVD, tackling a binary classification problem. There is still a lot of effort to be made in the future, so DL methods are able to classify the type of cardiovascular disease with the detail provided in Section 2.1. Accurately categorizing the type of CVD is crucial for efficient clinical treatment.

## Figures and Tables

**Figure 2 diagnostics-13-00068-f002:**
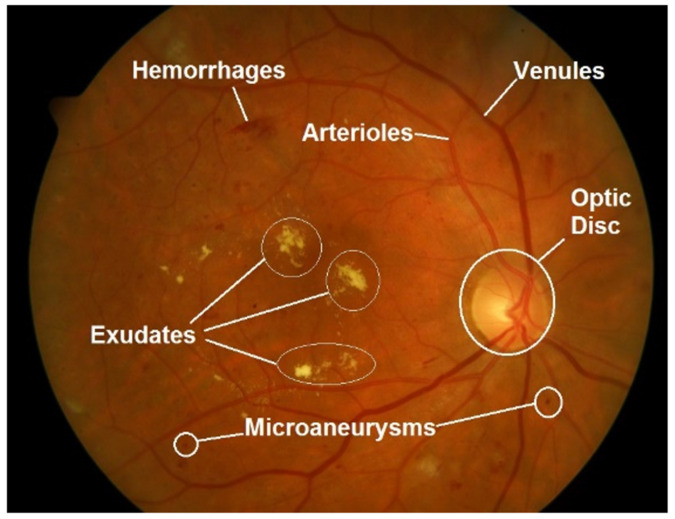
Retinal fundus image with marked main retinal features and abnormalities, typical signs of diabetic retinopathy (DR). Image source [21].

**Figure 3 diagnostics-13-00068-f003:**
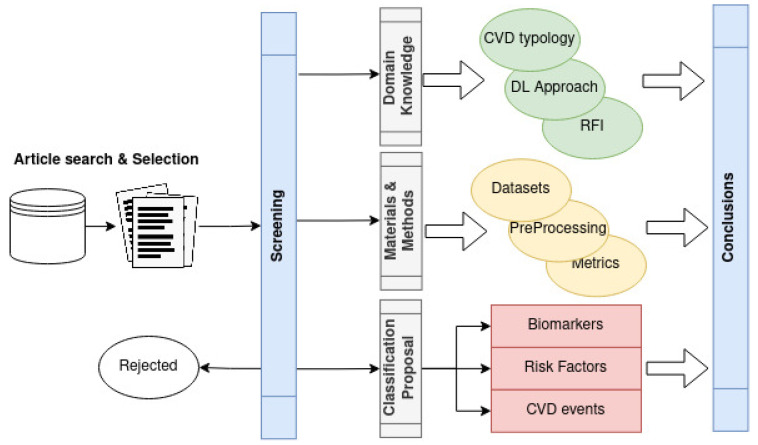
Proposed methodology overview.

**Figure 4 diagnostics-13-00068-f004:**
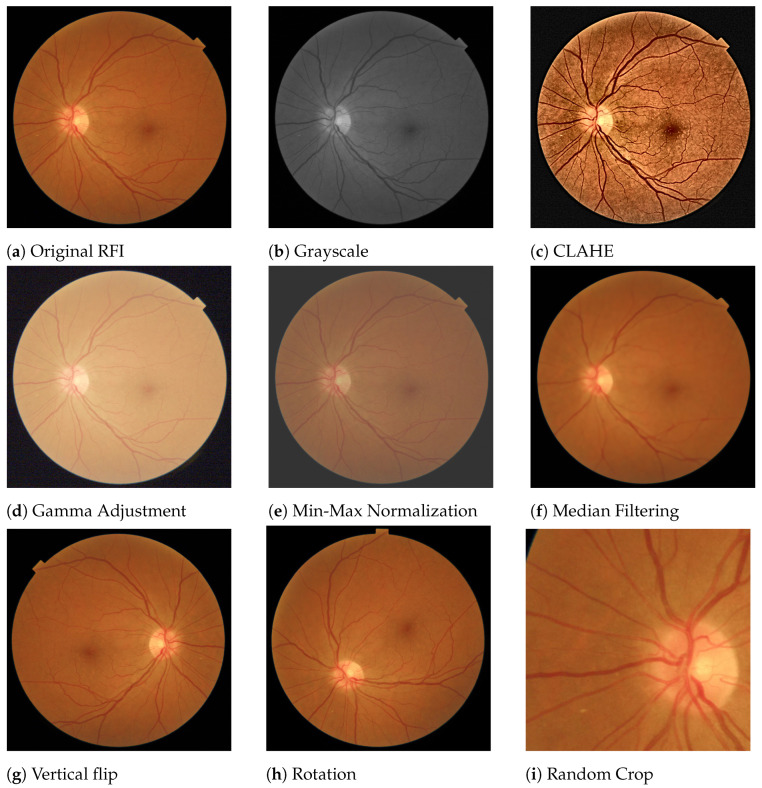
Pre-processing techniques commonly used in retinal images used in cardiovascular disease applications. (**a**) Original retinal image. (**b**) Color transformation: grayscale. (**c**) Contrast-Limited Adaptive Histogram Equalization. (**d**) Gamma correction with γ=2.5. (**e**) Min-Max normalization with [α,β]=[50,200]. (**f**) Noise removal: median filtering with N=29. (**g**–**i**) Data augmentation examples: Vertical flip, 45° rotation and random crop, respectively.

**Table 1 diagnostics-13-00068-t001:** Overview of public/restricted access retinal image datasets used in cardiovascular disease applications.

Name	#Images	Size	Labels	Application
**Public Access**				
DRIVE [22]	40	768×584	33—Normal, 7—Mild early DR	Vessel segmentation, DR diagnosis
Messidor-1 [23]	1200	1440×960, 2240×1488, 2304×1536		Macular Edema, DR diagnosis
Messidor-2 [24]	1748	1440×960, 2240×1488, 2304×1536	0-None, 1-Mild, 2-Moderate, 3-Severe, 4-Proliferative	DR prediction
STARE [25]	400	700×605		Vessel segmentation
HRF [26]	45	3504×2336	15—healthy, 15— DR, 15—glaucoma	Vessel segmentation, DR diagnosis, Glaucoma assessment
Kaggle/EyePACS [27]	9963		DR scale of 0–4	DR diagnosis
**Restricted Access**				
E-Optha [28]	463	2544×1696, 1440×960, 2048×1300	7—exudates and 35—no lesion 148—microaneurysms/ small hemorrhages and 233—no lesion	DR diagnosis
SEED [29]	235		43—Glaucomatous	Glaucoma assessment
SiNDI [30]	5783		5670—Healthy, 113—Glaucomatous	Glaucoma assessment
SCES [31]	1751			CVD assessment
SiMES [32]	1488			CVD assessment
BES [33]	8585			CVD assessment

**Table 2 diagnostics-13-00068-t002:** Overview of evaluation metrics used in cardiovascular disease applications. TP stands for true positives, FP stands for false positives, FN stands for false negatives and TN for true negatives.

Metric	Formula	Description
Accuracy	$Acc=\frac{\left(TP+TN\right)}{\left(TP+FP+TN+FN\right)}$	Indicates the global ratio of correct predictions either positive or negative
Precision/PPV	$Pr=\frac{TP}{\left(TP+FP\right)}$	Also known as Positive Predictive Value. Is the average of the retrieved samples that were relevant
Sensitivity/Recall/TPR	$Se=\frac{TP}{\left(TP+FN\right)}$	Also know as True Positive Ratio, indicates the ratio of the relevant samples that are successfully identified.
Specificity/FPR	$Sp=\frac{TN}{\left(TN+FP\right)}$	Also know as False Positive Ratio, is the ratio of identified negative samples that are actually negative.
F1-measure	$F1=\frac{2TP}{2TP+FP+FN}$	Represents the harmonic mean of the precision and recall
AUROC/AUC		Area Under Receiver Operating Characteristic. Relates TPR against FPR. It depicts the prediction capability of a classifier system as its discrimination threshold is varied
AUPRC		Area Under Precision-Recall Curve. Relates Precision and Recall providing a single number that summarizes the information in the PR curve
Sørensen–Dice coefficient	$DSC=\frac{2|X\cup Y}{\left|X|+|Y\right|}$	The Sørensen–Dice coefficient, or simply Dice score index, is a statistic metric used to compare the similarity of two samples
R²		The coefficient of determination, or R squared, is a statistical variable that represents the ratio of the variation in the dependent variable that is predictable from the independent variable(s)
CRAE		Center Retinal Venular Equivalent. Expressed in μm
CRVE		Central Retinal Arteriolar Equivalent. Expressed in μm

**Table 3 diagnostics-13-00068-t003:** Retinal vascular parameters and their influence in target end-organ damage and CVD [36].

Retinal Parameter		Change	Outcome
Tortuosity	Retinal arteriolar tortuosity	Increased	Current blood pressure and early kidney disease
		Decreased	Current blood pressure and ischemic heart disease
	Retinal venular tortuosity	Increased	Current blood pressure
Fractal dimension (FD)	Retinal vascular FD	Increased	Acute lacunar stroke
		Decreased	Current blood pressure, lacunar and incident stroke
		Suboptimal	Chronic kidney disease and coronary heart disease
Bifurcation	Retinal arteriolar branching angle	Decreased	Current blood pressure
	Retinal arteriolar branching asymmetry ratio	Increased	Current blood pressure
	Retinal arteriolar length: diameter ratio	Increased	Current blood pressure, hypertension and stroke
	Retinal arteriolar branching coefficient (optimal ratio)	Increased	ischemic heart disease
	Retinal arteriolar optimal parameter (deviation of junction exponent)	Decreased	Peripheral vascular disease.

**Table 4 diagnostics-13-00068-t004:** Automated methods for extracting biomarker characteristics.

Ref.	Application	Architectures	Metrics	Result	Dataset	Pre-Processing
[37]	Vessel segmentation	U-net	AUC	>0.90 (AUC)	UK Biobank (420), 21 datasets (4015), but filtering images with labels: Macular edema, Hypertensive, pathologic myopia and DR	CLAHE
[39]	Vessel segmentation	SC-GAN and U-Net	Acc, Precision, recall, Dice score	0.953 (Acc)	DRIVE	Grayscale, Median-filtering, CLAHE
[41]	Vessel segmentation	Custom U-Net-based	AUC, Acc, sensitivity, specificity	0.98 (AUC)	DRIVE, STARE	
[42]	Retinal vessel detection	U-Net	AUC, Acc (DICE score)	0.99 (AUC) in FD, 0.88 (AUC) in vascular density	UK BioBank (97895)	
[43]	Vessel detection	Fast RCNN	Sensitivity, Positive Predictive Value (PPV)	0.92 (sensitivity)	HRF, DRIVE, STARE, MESSIDOR, (450)	CLAHE

**Table 5 diagnostics-13-00068-t005:** Automated prediction of cardiovascular risk factors characteristics.

Ref.	Application	Architectures	Metrics	Result	Dataset	Pre-Processing
[46]	DR, Macular edema prediction	InceptionV3	AUC, sensitivity and specificity	0.99 (AUC)	EyePACS (128175): train, EyePACS (9963) and Messidor-2 (1748): test	
[48]	DR screening	Custom +Decision Tree	AUC, sensitivity and specificity	0.95 (AUC)	Total EyePACS MESSIDOR, E-Optha, (75137 in total)	Crop, brightness and contrast adjustment, DA: rotations
[50]	DR assessment	Custom CNN	Acc	0.89 (Acc)	Custom (150)	Min–Max normalization
[51]	DR, Glaucoma, AMD	Adapted VGG	AUC, sensitivity and specificity	0.95 (AUC), 0.94 (AUC), 0.93 (AUC)	Several datasets with more than 300,000 RFI just for train	
[52]	DR, Glaucoma, AMD	VGG19	AUC, Sensitivity and Specificity	0.93 (AUC)	10 diff studies: (76,730): train, (112648): test	
[53]	DR assessment	U-Net, ResNet101	AUC	0.93 (AUC)	IRIS	Grayscale, CLAHE, Gamma-Adjustment
[55]	Retinopathy of prematurity detection	U-Net	AUC, sensitivity and specificity	0.96 (AUC)	i-ROP Study (4861)	
[57]	CVD diagnosis	Inception-v3 + Ensembling of 10 iterations	AUC	0.70 (AUC)	Biobank (48,101), EyePACs (23,6234): train, Bionbank (12,026), EyePacs (999): test	
[58]	Biomarker prediction	MobileNet-V2	MAE, AUC, $R^{2}$	0.97 (AUC), 0.78 (AUC)	Qatar Biobank (12,000)	Gaussian filter, Crop, DA: flip, random rotation, shift
[61]	CVD risk, blood presure, body-mass index, total cholesterol, and glycated-hemoglobin level.	SIVA-DLS	$CRAE_{c}$, CRVE		SEED (Singapore Epidemiology of Eye Disease), BES (Beijing Eye Study), UK Biobank, Kangbuk Samsung Health (KSH), The Austin Health Study (Austin study).	
[62]	47 biomarkers prediction	VGG16	AUC, $R^{2}$	0.90 (AUC)	Two health screening centers in South Korea, the BES, SEED, UK Biobank (236,257)	CLAHE, DA: random crop, flip up-down, rotation, brightness, and saturation
[63]	BA prediction	VGG-	c-index, sensitivity, specificity	0.76 (sensitivity)	KHS (129,236): train, UK Biobank: test	
[64]	Retinal Age prediction	Xception	*p*-value and MSA	0.80 (*p* < 0.001), 3.55 (MSA)	UK Biobank (19,200)	
[68]	CAC assessment	InceptionV3	AUC	0.83(AUC)	Seoul National University Bundang Hospital (44,184)	Crop DA: flip, rotation
[69]	CAC assessment	Custom+ EfficientNet	AUC	0.74 (AUC)	Biobank, Shouth Korean Datasets (216,152)	CLAHE, DA: random crop, random rotation
[70]	CAC assessment	VGG16	Acc, Rec, Pre, F1, CM	0.78 (Acc), 0.75 (Rec), 0.91 (Pre)	Endocrinology Department, Vall d’Hebron University Hospital (152)	Crop
[72]	Proxys: hypertension, hyperglycemia and dyslipidemia	InceptionV3	AUC, Acc	0.76 (AUC), 0.88 (AUC), 0.70 (AUC)	China dataset (1222)	DA in minority classes
[73]	Hypertension prediction	DNFN	Acc, Rec, Pre	0.91(Acc)	fundusimage1000	Grayscale
[74]	Atherosclerosis assessment	Xception	AUROC, AUPRC, accuracy, sensitivity, specificity, positive and negative predictive values	0.71 (AUC)	Health Promotion Center of Seoul National University Hospital (15,408)	DA: Random zoom, horizontal flip
[75]	Coronary Artery Disease prediction	GraphSAGE	Acc, Rec, Pre, AUC	0.69 (AUC)	Custom	

**Table 6 diagnostics-13-00068-t006:** Automated prediction of cardiovascular events characteristics.

Ref.	Application	Architectures	Metrics	Result	Dataset	Pre-Processing
[76]	Stroke risk prediction	U-Net, VGG19	AUC, sensitivity and specificity	0.96 (AUC)	Singapore, Sydney and Melbourne and SCES, SiMES, SiNDI, DMPMelb, SP2	DA: random flipping and rotation
[77]	ICVD Diagnosis	Inception-ResNet-V2	AUC and $R^{2}$	0.97 (AUC)	China and Beijing Research on Aging, BRAVE (411518)	
[79]	CVD diagnosis	InceptionV3	Acc, F1	0.91 (Acc)	Biobank and EyePACS, STARE	
[80]	CVD diagnosis	CNN-ResNet+Custom	Acc, Rec, Pre	0.75 (Acc)	Qatar BioBank (1805)	Crop, Mean-filtering
[81]	Myocardial infarction prediction	Multichannel variational autoencoder, CNN-ResNet50	AUC, Sensitivity and Specificity	0.80 (AUC)	UK Biobank (71515)	

## Data Availability

Not applicable.

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
