# Peer review of "An Overview of Deep-Learning-Based Methods for Cardiovascular Risk Assessment with Retinal Images"

_diagnostics, 2022, doi:10.3390/diagnostics13010068_

Round 1
Reviewer 1 Report
Please see the attachment

Author Response
Notes to reviewer attached in document.

Reviewer 2 Report
I usually give detailed suggestions for the improvement of the ms. However this ms is very well written and sounds interesting as it is without the need of further improvements.
Author Response
Dear reviewer,
We would like to thank you for the positive comments made and for taking the necessary time and effort to review the manuscript.
Reviewer 3 Report
This is an interesting overview of the recent progress in deep-learning (DL) strategies to help diagnose cardiovascular diseases (CVDs) by virtue of retinal fundus images (RFIs). The authors introduce the domain knowledge, relevant materials and method in great detail, and review automated diagnosis of CVDs which are subdivided into 3 subsets according to the different focus of DL approaches. However, there are still some issues needed to be addressed before the manuscript is ready for publication.
Introduction
1. Line 34, “trough” is wrongly spelt, and it should be “through”.
Section 2
2. Line 105 and 116, “as” is missing after “such”, please change “such” to “such as”.
Section 4
3. Table 3 integrates automated methods for extracting biomarkers, mainly vessel segmentation and detection. How about the relationship between these biomarkers and cardiac events? Please add a column to Table 3 in order to summarize it.
4. There is something wrong with the way the author cites the references throughout the article, such as Line 266, “Also based in U-Net architecture, [44] proposes a method …” Generally speaking, “[number]” are usually placed in a complete sentence or at the end of the sentence. However, the authors use “[number]” as the pronoun, leaving the other parts of the sentence incomplete, which may cause ambiguity. How about changing “[number]” to “This paper”?
5. In 4.2, the authors outline several works which applied DL models to analyze risk factors of CVDs, such as diabetes, age, premature factors, etc. However, in this part, the authors do not directly elaborate on the relationship between risk factors and CVDs. I hope that the explanation of this part can be added so as to be more conducive to clinical application.
Conclusions
6. The authors outline a list of CVDs in section 2. Nevertheless, there is still a long way to go before these DL strategies are actually applied to the clinic and participate in the diagnosis and differentiation of different CVDs. The authors should expand on the limitations of the application of DL strategies, and propose your opinions about this issue.
Author Response
Notes to reviewer attached in document.

Reviewer 4 Report
The paper well reviewed the recent advances of the oculomics fields related to cardiovascular risk assessment.
1. Introduction – Recently, were also introduced in the oculomics fields. For example, musculoskeletal diseases and brain damage have been introduced. Please see the following articles: "Oculomics for sarcopenia prediction: a machine learning approach toward predictive, preventive, and personalized medicine" and "Emerging oculomics based diagnostic technologies for traumatic brain injury".
2. In fact, Figure 2 is not directly associated with cardiovascular diseases. The features are the signs of diabetic retinopathy (DMR).
3. The datasets shown in Table 1 don’t provide cardiovascular risks. Most of tehm are datasets for DMR.
4. The most important retinal features associated with cardiovascular diseases are vessel changes. The authors should describe the pathophysiology and detailed signs of the retina with a high cardiovascular risk.
5. Too many studies are found when diabetic retinopathy is included in the review. I recommend the authors should focus on “cardiovascular diseases”.
6. The coronary artery calcium score (CACS) has a very poor ability to discriminate between those who will and those who will not experience a clinical cardiovascular event (Coronary Artery Calcium Score and Cardiovascular Event Prediction | JAMA | JAMA Network ). However, many studies have tried to predict CACS. The authors discuss about this issue.
7. It is necessary to explain more organically how the algorithms are used to diagnose cardiovascular diseases using fundus images more accurately. For example, there is no explanation about the adavantages of the preprocessing techniques in Fig 3 to improve the cardiovascular risk assessments.
Author Response
Notes to reviewer attached in document.

Round 2
Reviewer 1 Report
Accepted in its present form
Reviewer 4 Report
The manuscript has been revised but the citations should be revised. All numbers are broken.